# A cellular mechanism for inverse effectiveness in multisensory integration

Torrey LS Truszkowski[1], Oscar A Carrillo[1], Julia Bleier[1],
Carolina M Ramirez-Vizcarrondo[1], Daniel L Felch[1], Molly McQuillan[2],
Christopher P Truszkowski[3], Arseny S Khakhalin[2], Carlos D Aizenman[1]*

[1]Department of Neuroscience, Brown University, Providence, United States; [2]Bard College, Annandale-On-Hudson, Unied States; [3]Roger Williams University, Bristol, United States

**Abstract** To build a coherent view of the external world, an organism needs to integrate multiple types of sensory information from different sources, a process known as multisensory integration (MSI). Previously, we showed that the temporal dependence of MSI in the optic tectum of *Xenopus laevis* tadpoles is mediated by the network dynamics of the recruitment of local inhibition by sensory input (Felch et al., 2016). This was one of the first cellular-level mechanisms described for MSI. Here, we expand this cellular level view of MSI by focusing on the principle of inverse effectiveness, another central feature of MSI stating that the amount of multisensory enhancement observed inversely depends on the size of unisensory responses. We show that non-linear summation of crossmodal synaptic responses, mediated by NMDA-type glutamate receptor (NMDARs) activation, form the cellular basis for inverse effectiveness, both at the cellular and behavioral levels.

*For correspondence:
Carlos_Aizenman@brown.edu

**Competing interests:** The authors declare that no competing interests exist.

## Introduction

Inverse effectiveness allows the brain to preferentially enhance multimodal input with low saliency, as compared to high saliency, readily detectable inputs (*Stein et al., 2009*). The phenomenological aspects of inverse effectiveness and other properties of MSI are well described in the vertebrate optic tectum or superior colliculus (*Stein and Stanford, 2008*; *Stein et al., 2014*; *Meredith and Stein, 1983*, *1986*), but less is known about the cellular mechanisms underlying these processes or how these responses give rise to behavior (*Stein et al., 1988*, *1989*). While several cellular models have been put forth to explain inverse effectiveness (*Cuppini et al., 2012*; *Ursino et al., 2014*; *Stein et al., 2009*), one stumbling block toward testing these has been the lack of a robust, experimentally tractable model system that is easily assessable at multiple levels of analysis, from synapses to behavior. The *Xenopus laevis* tadpole optic tectum has emerged as a preparation in which we can study MSI at the single cell, network and behavioral levels (*Deeg et al., 2009*; *Felch et al., 2016*). The optic tectum receives synaptic input from multiple sensory modalities, particularly visual, auditory and mechanosensory projections and shows strong MSI with properties homologous to those observed in the mammalian superior colliculus (*Deeg et al., 2009*; *Hiramoto and Cline, 2009*).

## Results and discussion

Our recent findings (*Felch et al., 2016*) showed that the basic properties of MSI in the superior colliculus are also present in the *Xenopus* tadpole optic tectum which provides an experimentally tractable model system for studying the cellular basis of MSI. Critically, these findings show that inhibition is required for the development of the temporal properties of MSI in single cells.

However, this study left several open questions, including whether other properties of MSI, such as inverse effectiveness, are also also present in this system; and whether MSI measured in single tectal neurons also reflects MSI in vivo and in a behaving animal. The present study extends our previous findings by directly addressing these questions. Specifically, we focused on whether inverse effectiveness is also a property of MSI in the tectum as assessed by multiple levels of analysis. We use this integrated preparation to examine the behavioral, network and cellular mechanisms underlying inverse effectiveness.

Individual tectal cells show robust MSI as measured with extracellular single-cell recordings (*Felch et al., 2016*). Here, we first tested whether MSI was also present in tadpole behavioral responses, and whether this behavior was consistent with inverse effectiveness. In *Xenopus* tadpoles, visually guided behavior and acoustically driven startles are well characterized, and ideal for testing MSI (*Figure 1A*) (*Khakhalin et al., 2014*; *James et al., 2015*; *Truszkowski et al., 2016*). Tadpoles change their swimming speed (in cm/s) when presented with a visual counterfacing grating in a manner directly proportional to the contrast of the grating (*Figure 1B*, Visual: 0%: 3.22 ± 0.25; 25%: 3.29 ± 0.36; 50%: 3.7 ± 0.51, 100%: 4.76 ± 0.53) (*Schwartz et al., 2011*). Visual stimuli were paired with an acoustic prestimulus that by itself does not evoke a startle response. Pairing the visual stimulus with a subthreshold acoustic prestimulus enhanced the change in speed only when paired with low-contrast visual stimuli and not with high-contrast stimuli (*Figure 1B*, Multisensory: 0%: 3.22 ± 0.27, p>0.9999; 25%: 5.19 ± 0.62 cm/s, p=0.0013; 50%: 3.88 ± 0.51 cm/s, p=0.9948; 100%: 4.03 ± 0.34, p=0.4994, two way ANOVA). We also quantified this difference as a Multisensory Index [MSIn = (paired response – visual response) / visual response]. At 25% and 50% contrast, the MSIn was significantly different from the unisensory or 0% stimulus, but not for 100% contrast (*Figure 1C*, MSIn: 0%: 0.018 ± 0.05; 25%: 0.92 ± 0.244, p=0.002; 50%: 0.68 ± 0.25, p=0.0345; 100%: 0.28 ± 0.18, p=0.3981; two-way ANOVA). Taken together, these data show that the tadpoles use acoustic information to enhance behavioral responses only to low saliency visual stimuli but not to high saliency visual stimuli, showing a behavioral correlate of inverse effectiveness.

We next tested whether individual tectal neurons expressed inverse effectiveness to physiological stimuli. We used tectum-wide calcium imaging to observe network responses to paired sub-maximal visual and mechanosensory stimuli, in vivo. Tectal cells were bulk-labeled with Oregon Green BAPTA 1 (OGB1-AM) to measure responses from up to 170 tectal cells simultaneously (*Xu et al., 2011*). Visual stimuli were presented via an optic fiber 400–600 µM from the eye and skin-based sensory (presumed mechanosensory) stimuli were presented via a bipolar stimulating electrode on the skin in the lip area. Recordings were obtained from the contralateral optic tectum. Each cell was assigned a primary modality (visual, mechanosensory or multisensory) based on its largest average peak response (*Figure 1D*). In the representative example shown, 51% of cells are primarily multisensory, 28% are mechanosensory and 13% are visual. Nine percent do not respond to any stimulus. Larger circles represent proportionally larger responses. Example responses from individual regions of interest (ROI) representing single cells are shown (*Figure 1E*). Combining data from 11 tadpoles with 35–170 ROIs each (1041 ROI total; 231 with average peak response to at least one stimulus greater than $\Delta F = 0.1$), we calculated the average peak response across each modality and the MSIn. Our data shows that inverse effectiveness is met across the population of tectal cells, with cells having small unisensory responses showing the greatest amount of multisensory enhancement (*Figure 1F*). Because inverse effectiveness was observed across a network of cells in response to a single stimulus of a given strength, these data provide direct experimental evidence supporting the definition of inverse effectiveness as a function of an individual cell's response size to a stimulus, rather than a function of stimulus strength.

What are the cellular mechanisms that mediate inverse effectiveness? To better understand how cellular properties can give rise to the network and behavioral responses observed above, we performed a series of single-cell recordings. Previously, we found that inhibition is crucial for developing the temporal properties of MSI. Inverse effectiveness could also be explained by enhanced recruitment of local inhibition by increasingly large unisensory stimuli. Using an ex vivo preparation (see Materias and methods [*Felch et al., 2016*]) to independently stimulate each modality with precise control of stimulus magnitude and timing, we examined the relationship between inverse effectiveness and inhibition. We found that inverse effectiveness is not altered by blocking $GABA_A$ receptors (*Figure 2A*, 100 µM PTX). Although in the presence of GABA blockers, the response curve is shifted

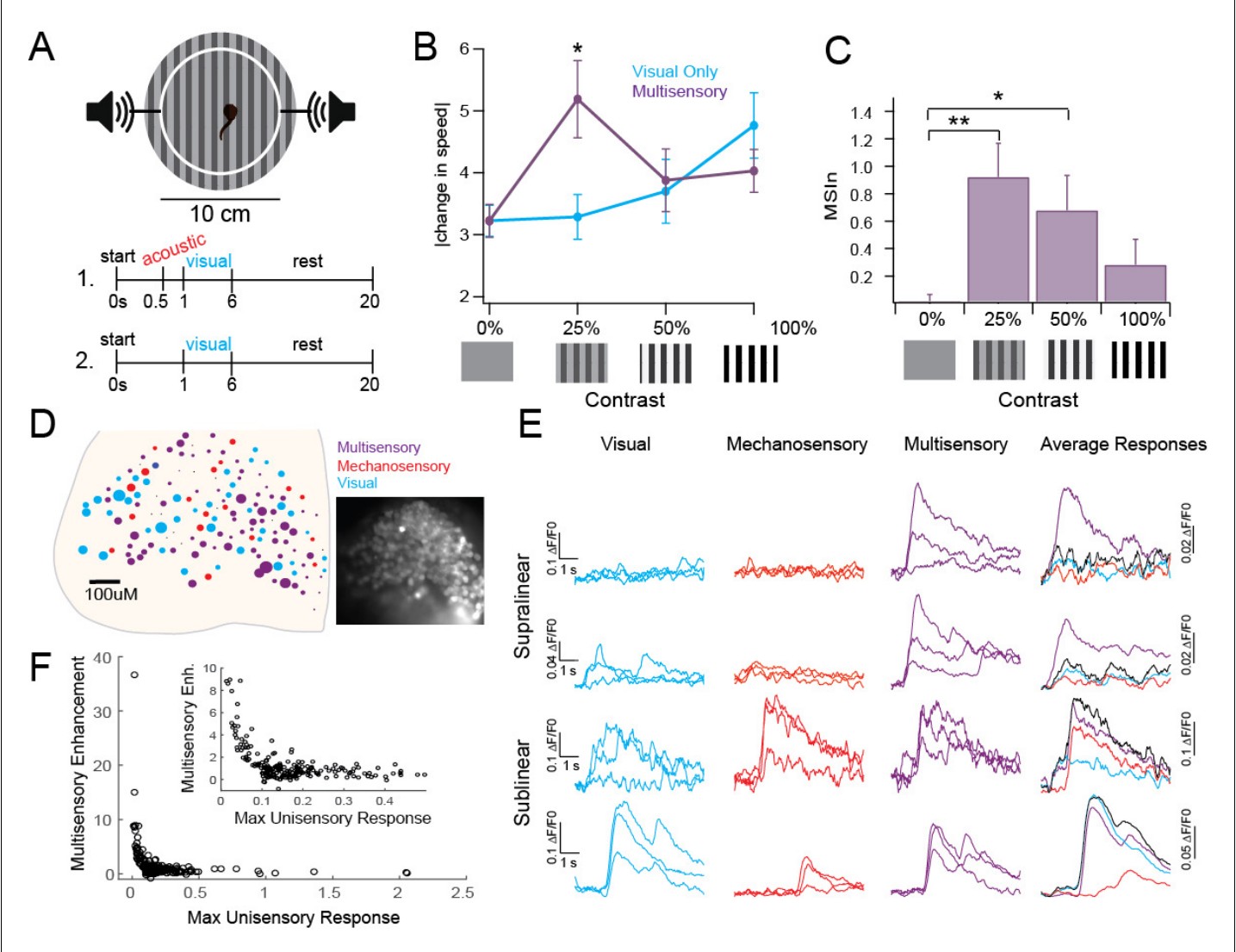

**Figure 1.** Inverse effectiveness persists at behavioral and network levels. (**A**) Diagram of behavioral experiment showing timing of auditory and visual stimulus delivery, see Methods for details. (**B**). Change in swimming speed in response to visual stimuli of varying contrasts, with and without subthreshold acoustic prestimulus. Notice that multisensory enhancement is only observed with low saliency stimuli. Mean and SD are plotted. Unisensory: 0%: 3.22 ± 0.25, 25%: 3.29 ± 0.36 cm/s, 50%: 3.7 ± 0.51, 100%: 4.76 ± 0.53 cm/s. Multisensory: 0%: 3.22 ± 0.27, 25%: 5.19 ± 0.62 cm/s, 50%: 3.88 ± 0.51 cm/s, 100%: 4.03 ± 0.34 cm/s. n = 37 tadpoles each receiving visual and multisensory stimuli across every randomly assigned contrast. Two-way analyses for Visual Condition Vs. Multisensory condition: 0%: p>0.999, t(144)=0.005431; 25%: p=0.0013, t(144)=3.684; 50%: p=0.9948, t(144)=0.3431; 100%: p=0.4994, t(144)=1.416. (**C**) MSIn for each contrast level. All individual responses, as well as mean and interquartile range are plotted. Mean and SD: 0%: 0.018 ± 0.05, 25%: 0.92 ± 0.244, 50%: 0.68 ± 0.25, 100%: 0.28 ± 0.18. n = 37 tadpoles as described in *Figure 1B* legend. One-way analyses for MSIn: 0% Vs. 25%: p=0.002, q(36)=3.703; 0% Vs. 50%: p=0.0345, q(36)=2.613; 0% Vs. 100%: p=0.3981, q(36)=1.357. (**D**) Distribution of cells across one tectum coded by response type and intensity. Color indicates primary modality; size indicates size of largest response. Inset: Fluorescent image of OGB1-AM loaded tectum. (**E**) Example responses to unisensory and multisensory stimuli indicating different types of multisensory interactions. (**F**) Plot of single modality response vs MSIn across 231 ROIs shows IE. Exponential fit = $a^{(b*x)}$; a = 27.21 (21.93, 32.5), b = −41.03 (−47.85,−34.21), Adjusted $R^2$ = 0.575.

The following source data is available for figure 1:

**Source data 1.** Data for *Figure 1*.

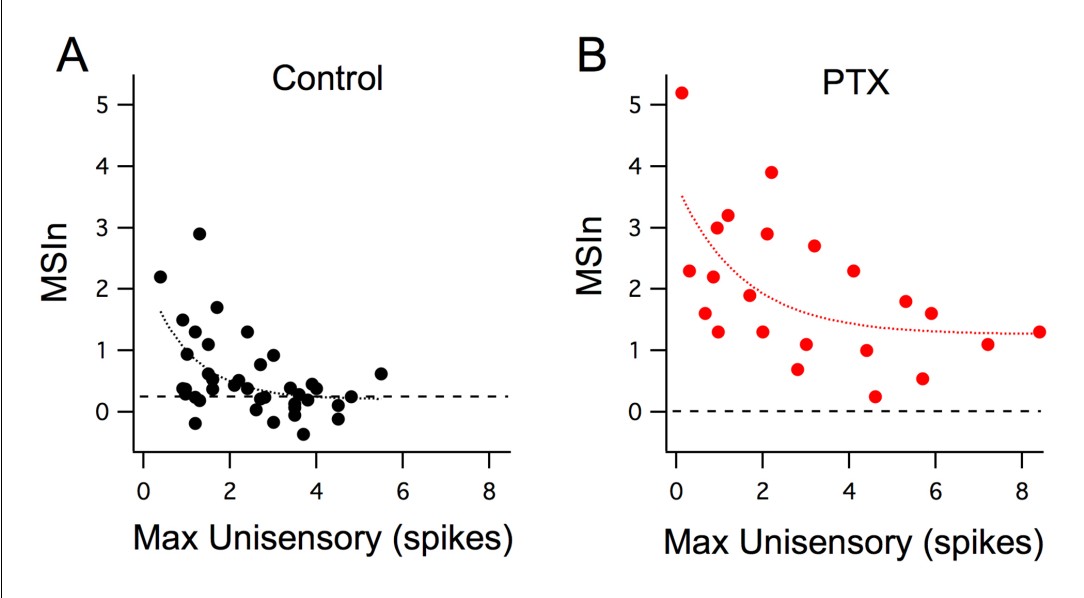

**Figure 2.** Inverse effectiveness is not dependent on inhibition. (**A**) Maximum unisensory responses in spike output plotted against MSIn in Stage 49 tadpoles in control (n = 40, cells) conditions and with GABA-R blocker (n = 22; picrotoxin). Despite the apparent boost in MSI for the picrotoxin group across all response sizes, the decay trend for inverse effectiveness remains intact and similar to the control group. Curves represent a single exponential decay fit using the least-squares method.

The following source data is available for figure 2:

**Source data 1.** Data for *Figure 2*.

to greater unisensory and multisensory responses, smaller unisensory responses are still associated with larger multisensory enhancement, and thus inverse effectiveness is still present.

A second hypothesis is that inverse effectiveness may arise from the supra-linear summation of responses to multisensory stimuli in individual tectal neurons that results from active dendritic properties. To test this, we measured evoked synaptic responses to supra and subthreshold, unisensory and multisensory stimuli using whole-cell current clamp recordings (*Figure 3A,B*). We observed that tectal cells exhibiting suprathreshold unisensory responses showed no multisensory enhancement resulting from crossmodal stimulus pairs. In contrast, tectal cells with subthreshold unisensory responses revealed a large multisensory enhancement of the combined response (*Figure 3B*, subthreshold MSIn: 4.34 ± 1.176, n = 10 suprathreshold MSIn: 0.12 ± 0.035, n = 6, p=0.0002, Mann-Whitney U = 0). This suggests that a non-linear process underlies summation of multisensory inputs in tectal neurons.

One possible mechanism for this non-linear summation involves recruitment of NMDA receptors (NMDARs), which are known to be coincidence detectors that can mediate non-linear dendritic integration properties (*Binns and Salt, 1996*). For example, individual small synaptic inputs alone may not be sufficiently strong to recruit NMDAR, but paired inputs will, and thus result in a response greater than the sum of the individual inputs (*Polsky et al., 2004*). To test whether NMDA receptor activation could explain the supralinear integration of subthreshold crossmodal input pairs in the tectum, we compared the arithmetic linear sum of both unisensory responses to subthreshold stimuli, to the actual combined multisensory response. This was then used to calculate the MSIn (see Materials and methods, *Figure 3C*). We observed that in NMDAR-blocked cells, multisensory enhancement diminishes (*Figure 3C,D*: Sub-threshold control MSIn = 4.34 ± 1.18, n = 6; Sub-threshold NMDAR-block MSIn = 1.11 ± 0.43, n = 10; p=0.016; Mann Whitney U = 8), and the multisensory response is closer to the linear sum of the individual responses (*Figure 3E*). To compare how this effect in whole-cell responses relates to the output of the cell, we used cell-attached recordings to measure spike output with and without NMDARs blocked. As with whole-cell responses, lower

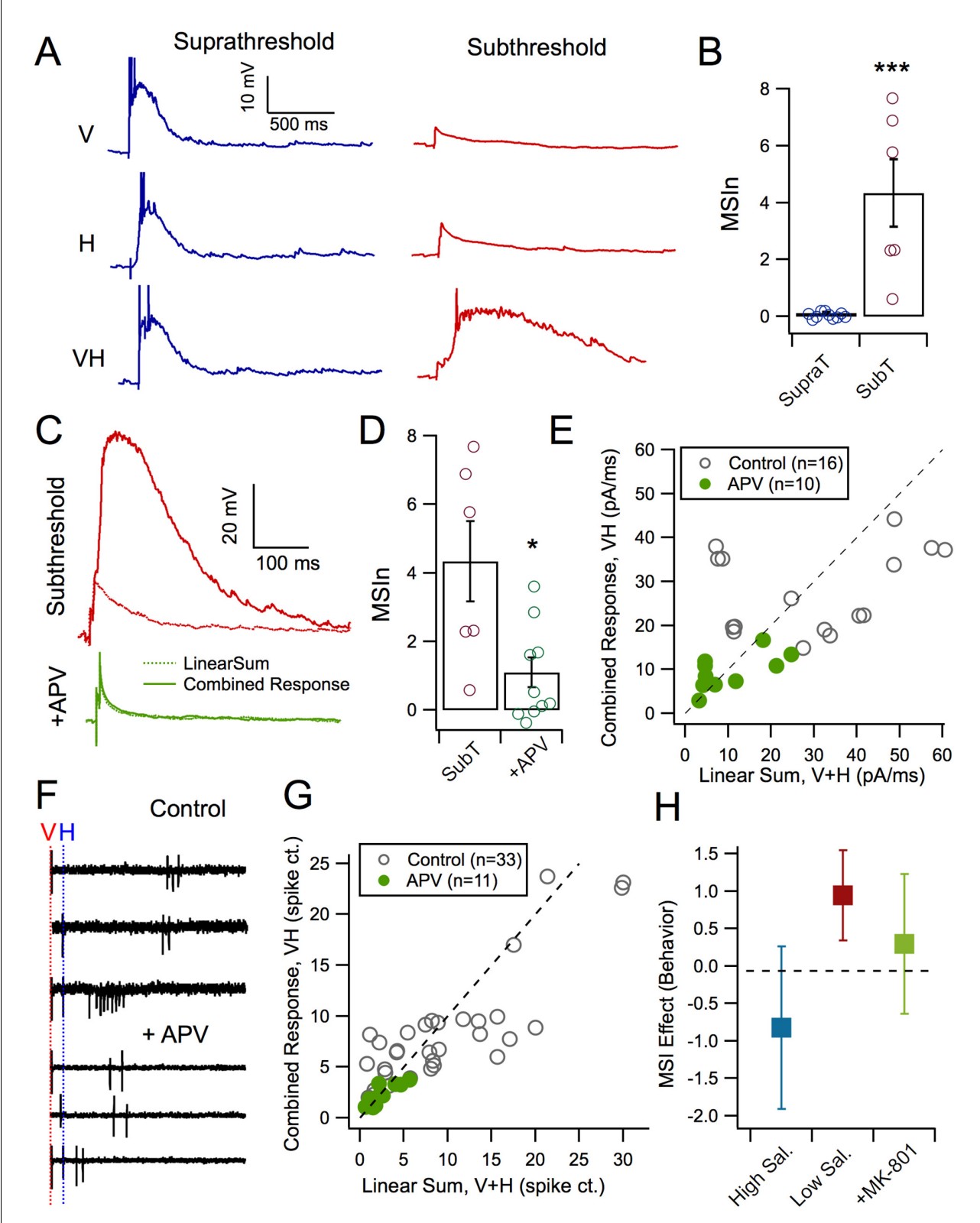

**Figure 3.** NMDAR activation mediates MSI. (**A**) Evoked synaptic responses produced either by a visual stimulus only, a mechanosensory stimulus only, or paired stimuli with a 50 ms interval. Pairing small, subthreshold responses results in large MSI. (**B**) Average MSIn for suprathreshold (MSIn = 0.12 ± 0.035, n = 10 cells) and subthreshold stimuli (MSIn = 4.34 ± 1.176, n = 6 cells, p=0.0002, Mann-Whitney U = 0). (**C**) Comparison of arithmetic sum of evoked subthreshold visual and hindbrain responses to evoked crossmodal responses in control and with NMDAR antagonist, APV

*Figure 3 continued on next page*

*Figure 3 continued*

(50 mM). (**D**) NMDAR-blocked cells exhibit significant lower levels of MSIn as compared to control cells (Control MSIn = 4.34 ± 1.18, n = 6 cells, NMDAR-block MSIn = 1.11 ± 0.43, n = 10 cells, p=0.016, Mann-Whitney U = 8). (**E**) Comparison of the linear sum of evoked visual and hindbrain responses (V+H) against the actual evoked crossmodal response (VH). Line indicates linearity. Values above and below diagonal show supralinear and sublinear multisensory responses, respectively. Notice that the APV group more closely approaches linearity. (**F**) Example loose-cell attached spike recordings to different stimulus conditions with and without APV. Note that NMDAR-blocked cells exhibit stunted supralinear multisensory responses. (**G**) Comparison of the linear sum of evoked visual and hindbrain spiking responses (V+H) against the actual evoked crossmodal response (VH). (**H**) Tadpoles in different experimental groups demonstrated different levels of behavioral MSI (ANCOVA $F_{(2,50)}$=4.1, p=0.02 after adjustment for tadpole responsiveness to unisensory acoustic stimuli: covariate $F_{(1,50)}$=6.1, p=0.02). In the pharmacological control group, MSI for low-contrast stimuli (0.12 ± 0.13; n = 15) was significantly higher than both zero (one-sample t-test p=0.005), and MSI for high-contrast visual stimuli (−0.10 ± 0.31, n = 23; post-ANCOVA Tukey HSD test p=0.03). The effect of MSI for low-contrast visual stimuli after pharmacological blockade of NMDA receptors with MK801 was close to zero (0.04 ± 0.21, n = 15), and was not significantly different from control MSI for either low- or high-contrast stimuli (Tukey HSD p>0.2).

The following source data is available for figure 3:

**Source data 1.** Source data for *Figure 3*.

intensity unisensory spike outputs in the presence of NMDAR-blockers do not elicit multisensory enhancement as observed in control conditions (*Figure 3F,G*: Control n = 33, NMDAR-Block n = 11).

Finally, we tested whether blocking NMDARs also prevented MSI as measured behaviorally. As before, multisensory behavior was compared with visual stimuli of high and low saliency in control tadpoles. Tadpoles treated with NMDAR blocker, MK-801 were also tested for MSI with low saliency stimuli. Tadpoles in different experimental groups demonstrated different levels of MSI (ANCOVA p=0.02 after adjustment for tadpole responsiveness to unisensory acoustic stimuli; covariate p=0.02). In the pharmacological control group, MSI for low-contrast stimuli (94 ± 109%; n = 15) was significantly higher than both zero (t-test p=0.005), and MSI for high-contrast visual stimuli (−82 ± 250%, n = 23; post-ANCOVA Tukey HSD test p=0.03). The effect of MSI for low-contrast visual stimuli after pharmacological blockade of NMDA receptors with MK801 was not significantly different from zero and was not significantly different from control MSI for either low- or high-contrast stimuli (29 ± 174%, n = 15; Tukey HSD p>0.2).

Taken together, our results show that inverse effectiveness persists at the single cell, network and behavioral levels, and relies on non-linear response integration in single tectal cells. These results extend those in our previous study to show that another important property of MSI is also present in the tectum, indicating that it is an evolutionarily important process conserved across vertebrates. These non-linear responses are mediated via NMDAR activation by co-incident subthreshold inputs. These data are the first to explicitly show the cellular mechanisms underlying inverse effectiveness in the vertebrate brain and may suggest a broader role for NMDARs in sensory perception.

## Materials and methods

All animal experiments were performed in accordance with and approved by Brown University Institutional Animal Care and Use Committee standards.

### Experimental animals

Wild-type *Xenopus Laevis* (RRID:NXR_0.0031) tadpoles were raised in Steinberg's rearing media on a 12 hr light/dark cycle at 18–21°C for 7–8 days, until they reached developmental stage 48 or 49, depending on the experiment (*Nieuwkoop and Faber, 1956*; *Truszkowski et al., 2016*). Developmental stages of tadpoles were determined according to Nieuwkoop and Faber (*Nieuwkoop and Faber, 1956*). The rearing medium was renewed every three days. At least two different clutches of tadpoles from different husbandry were used for every set of experiments. Sexual differentiation of tadpoles is not possible at these developmental stages.

## Behavior

### Multisensory behavior

During the light cycle, individual stage 48–49 tadpoles were placed in an 87 mm diameter dish filled with Steinberg's solution and exposed to a series of 40 block randomized stimulus presentations, half visual and half multisensory with a 20 s interstimulus interval (*Figure 1A*). Visual stimuli consisted of grayscale stripes of three contrasts (25%, 50%, 100%) and a control 0% condition (gray), which was the same as the background during the 20 s interstimulus interval. Stripes alternated at 4 hz for 2 s on a CRT monitor beneath the dish. Multisensory stimuli consisted of visual stimuli of each contrast preceded by 0–100 ms by an acoustic stimulus. Acoustic stimuli consisted of low-volume clicks played through a set of two speakers, connected to the dish on either side by wooden struts so that they vibrated the liquid in which tadpoles swam. The acoustic stimulus likely activates the ear, skin and lateral line sensory systems. The volume was calibrated to be sub-threshold, that is the acoustic stimulus alone did not elicit a startle response, using a prepulse inhibition protocol (*James et al., 2015*). Trials lasted approximately 16 min. Both visual and acoustic stimulus presentations were controlled by a MATLAB script to ensure precise timing.

For analysis, tadpoles that did not move in response to any stimulus for at least three consecutive minutes were eliminated. Tadpole velocity was averaged over 1 s pre-trial and during the trial, and the absolute value of the percent change was calculated. All trials from each condition were averaged, and the multisensory index (MSIn = (multisensory – unisensory) / unisensory) was calculated. Mean and standard error are plotted. Due to the relatively large sample size, a two-way ANOVA was used with Sidak's multiple comparisons test to compare visual and non-visual groups across the different contrasts. Additionally, a one-way ANOVA was used with Dunnett's multiple comparisons test to compare the MSIn at 0% contrast against 25, 50 and 100% contrast.

### NMDAR-blockade behavior

The behavioral apparatus consisted of a plastic Petri dish 10 cm in diameter, placed on an opaque acrylic screen, and surrounded from all sides with a cardboard box. The dish was connected to audio speakers (Arctic S111, Switzerland) with two wooden struts, while visual stimuli were projected on the screen from below using a video projector (AAXA P300, Aaxa Technologies, Tustin, CA). The dish was filled with rearing solution, 1.5 cm deep. Stage 48–49 tadpoles were placed in the dish one by one, and subjected to visual, acoustic, or multisensory stimulation (original script, written in P5 Java Script library), with stimuli of different modalities presented every 30 s, in a cycle. In the majority of experiments, this cycle consisted of: acoustic stimulus, low-contrast visual stimulus, acoustic stimulus and low-contrast visual stimuli combined, high-contrast visual stimulus, and acoustic stimulus combined with high-contrast visual stimulus. In some experiments, only low-contrast visual and multisensory stimuli were used. The acoustic stimulus consisted of one 10 ms pulse of a 100 Hz sawtooth buzz that was delivered to right and left speakers with inverted phases, to enable efficient sound and vibration transfer. The acoustic volume was calibrated to be just below the threshold for current batch of tadpoles (typically 7–10% of maximal volume). The visual stimulus consisted of a checkerboard pattern of light gray squares shown against a dark gray background, with 24 rows, each 6 mm wide, and light to dark gray contrast of either 10% (low), or 30% (high). The checkerboard pattern was not presented at once, but each light gray square within the pattern expanded over the course of 500 ms in such a way that one corner of each square stayed in place, while the opposite corner linearly translated to its final position. The direction of expansion was randomized between the trials. As this visual stimulus is not instantaneous, for multisensory presentations we was always started it 100 ms before the onset of the acoustic stimulus. The tadpoles were recorded from above using an HD USB video camera (C310, Logitech, Newark, CA); 30 s periods between stimuli were then automatically excised (Python script, courtesy of Alexander Hamme, Bard College), and startle responses were identified manually. The total number of stimuli delivered for each tadpole was either 25 (for experiments with five types of stimuli), or 30 (for experiments with three types of stimuli). Tadpoles that did not respond to any of the 25–30 stimuli were excluded from analysis.

We used ANCOVA and Tukey tests to analyze NMDAR-blockade effects on behavior, with 51 degrees of freedom used in this analysis (N−k = 51). Standard analysis of variance is appropriate because sample sizes are sufficiently large and raw data (the number of positive responses out of either 5 or 10 presentations) is distributed binomially. There was no randomization of stimuli as the

stimuli were provided cyclically (see above). Control and pharmacological condition experiments were run in different months (but otherwise equivalent batches of animals), and so were not randomized. The scoring criteria were formalized as a presence of a C-turn followed by a rapid acceleration. These behaviors are sufficiently robust to allow for unambiguous and internally consistent scoring. Trials were also independently scored by a second observer, cross-validating our scoring criteria. Means and 95% confidence levels are reported.

## Calcium imaging

### Calcium imaging preparation and recording

Stage 49 tadpoles were anesthetized in 0.02% tricainemethane sulfonate (MS-222; Sigma) and paralyzed in 0.12 mM Tubacurarine (Tocris bioscience (Bristol, UK) cat # 2820) for dissection. Dissection was completed in external solution (115 mM NaCl, 4 mM KCl, 5 mM HEPES, 10 µM glycine, 10 mM glucose). Dissection consisted of opening the skin above the brain, separating the two tecta and laying the contralateral tectum flat. Then, the ventricular membrane was removed from the recording area and 4 mM Oregon Green Bapta 1 (OGB1, Molecular Probes (Waltham, MA) cat # 06807) dye in 4% F-127 detergent was added and incubated in the dark for 1 hr (*Xu et al., 2011*). External media was changed with three intermediate rinses to remove excess dye, and then recording proceeded for 1–3 hr based on the health of the tadpole. Recordings were completed using an ANDOR 860 EM-CCD camera at 60x and NIS-elements software. Images were taken using $2 \times 2$ binning for an image size of $500 \times 502$ pixels, corresponding to 1130 µM x 1135 µM and 22.8 frames per second. Trials were presented in blocks of three presentations of four stimuli conditions (multisensory, visual, electrical stimulation of the skin (mechanosensory) and no stimulus), with each trial being 7 s long, with a 5 s pre-stimulus light stabilization time period and a 30 s start to start interval. Rest periods of at least 2 min were observed between trial blocks. The skin stimulation protocol is treated as functionally equivalent to presentation of acoustic stimuli (ie. non-visual) and so was used for convenience in order to minimize mechanical movement of the preparation induced by the stimulus during Ca++ imaging. Each tadpole generated 5–14 blocks of trials from 1–3 planes of focus for a total of 15–170 regions of interest per tadpole.

### Analysis of calcium imaging data

Data were extracted using a custom Matlab GUI (C Deister, github.com/cdeister). In brief, image files were converted to.tiff stacks and aligned in X-Y using a 100–300 frame average as a template. ROIs were hand selected and mean fluorescence of each ROI and its corresponding neuropil in each frame was extracted. The mean fluorescence was processed (custom Matlab code, TLST) to first remove background signal and account for bleaching, then neuropil was subtracted (*Chen et al., 2013*). The mean of the prestimulus period in each trial was used to calculate the change in fluorescence over time, and peak measurements were taken from these $\Delta F/F_0$ traces. Peak was calculated as the mean of the three data points around the absolute peak. The mean of the peak for all traces of a given condition for a given ROI were used to calculate primary modality and multisensory index (MSIn = (multisensory – unisensory) / unisensory). Cells with average ROI $\Delta F/F_o$ peak values less than 0.1 for all three conditions—visual, mechanosensory and multisensory—were excluded from analysis.

## Single-cell electrophysiology

For ex-vivo brain recordings, tadpole brains were prepared as described previously (*Pratt et al., 2008*; *Wu et al., 1996*; *Deeg et al., 2009*). In brief, tadpoles were anesthetized in MS-222 and then transferred to a HEPES-buffered extracellular saline solution (external solution). This solution contains (in mM) 115 NaCl, 2-4KCl, 3 CaCl2, 3 MgCl2, 5 HEPES, 10 glucose, and 0.1 picrotoxin (Sigma, St. Louis, MO). The solution was maintained at pH 7.2 and 255 mosM. For NMDAR-block experiments, 50 mM of D-(-)−2-Amino-5-phosphonopentanoic acid (AP5; Sigma) was added. In this external solution, we dissected the tadpole to expose the ventral surface of the optic tectum. This was accomplished by slicing through the dorsal aspect of the tadpole, penetrating first the pigmented skin and afterwards the dorsal brain through the midline. This process involveed cutting the membrane commissures and excising connective nerve fibers. The whole-brain preparation was then pinned to a submerged silicone elastomer (Sylgard, Dow Corning, Midland, MI) block in a custom-made recording rig. Broken micropipettes were used to suction and remove the ventricular

membrane overlying tectal cells. These tectal cells were resolved using a Nikon E600 FN light microscope (Tokyo) with a 60x fluorescent water-immersion objective. Synaptic responses were evoked using bipolar stimulating electrodes placed in the optic chiasm (visual) and the contralateral hindbrain (non-visual/mechanosensory). These stimuli represented a reduced version of the skin stimulation and acoustic stimuli presented in the behavior and calcium imaging experiments, but allow specific control of intensity. A 0.02 ms electrical shock was administered to each modality at various stimulus intensities to elicit a tectal response that was collected via either whole-cell patch clamp or cell-attached recording techniques. Multisensory stimuli were presented at a 50 ms offset interval. Stimulus intensity was set so as to evoke a monosynaptic tectal cell response with minimal confounding polysynaptic activity. Whole-cell and loose-cell attached recordings were performed using glass micropipettes (6–15 MΩ) filled with $K^+$-gluconate intracellular saline [in mM: 100 $K^+$-gluconate, 8 KCl, 5 NaCl, 1.5 MgCl2, 20 HEPES, 10 EGTA, 2 ATP, and 0.3 GTP, pH 7.2 (osmolarity 255 mOsm)]. Voltage-clamp mode was used to isolate synaptic conductances mediated by optic chiasm projections and hindbrain projections, but recordings were collected in current clamp conditions. Loose-cell attached recordings were used to measure action potentials without breaking through the cell membrane and without electrical access, and were defined as having seal resistances in the 40–200 MΩ range.

Action potentials were detected off-line by importing the digitized traces into the AxoGraphX analysis environment and by using a manually-determined amplitude threshold to identify events and determine post-stimulus onset times. The data were collected within 1.2 s after the initial stimulus onset. Whole-cell voltage recordings were also analyzed off-line by importing the digitized traces into the AxoGraphX analysis software. Responses were quantified by measuring the area under the voltage traces. Trials in which both modalities elicited areas each less than 12 pA/ms over the 1.5 s interval following the stimulus were labeled as 'Sub-threshold,' and trials in which either modality elicited a current area greater than 12 pA/ms over the 1.5 s interval following the stimulus were labeled as 'Supra-threshold.' Due to the limited sample size, non-parametric Mann-Whitney U-tests were performed using GraphPad Prism software to measure statistical significance. Means, standard errors of the mean and two-sided p-values are reported. Cells were held at −65 mV, and cells that maintained this voltage with currents greater than 30 pA were excluded from data collection and analysis. Additionally, cells that did not elicit consistent and reliable monosynaptic voltage-clamped responses for both visual and mechanosensory locations were excluded from data collection and analysis.

## Reagents

Reagents were obtained from Sigma (St. Louis, MO) unless otherwise indicated.

## Data availability

The data that support the findings of this study are available as a supplemental file.

## Code availability

All codes are available on Github. Custom calcium imaging data extraction programs were generously shared by Chris Deister (Brown University) and are available at https://github.com/cdeister/imageAnalysis_gui (*Deister, 2016*). Custom calcium imaging analysis programs and behavior analysis programs developed for this study are available at https://github.com/torreydactyl/Inverse-effectiveness-paper (*Truszkowski, 2017*; with a copy archived at https://github.com/elifesciences-publications/Inverse-effectiveness-paper). All code was generated in Matlab 2014 or later.

## Acknowledgements

We thank Mimi Oupravanh for animal care and experimental support. We thank the Cold Spring Harbor Laboratory Imaging Structure and Function in the Nervous System course leadership for technical assistance in the calcium imaging experiments. We thank Christopher Deister and Christopher Moore for assistance in analyzing the calcium imaging data.

## Additional information

### Funding

| Funder | Grant reference number | Author |
| --- | --- | --- |
| National Institutes of Health | NIH F31 NS09379001 | Torrey LS Truszkowski |
| National Science Foundation | NSF IOS 1353044 | Torrey LS Truszkowski<br>Oscar A Carrillo<br>Julia Bleier<br>Carolina M Ramirez-Vizcarrondo<br>Christopher P Truszkowski<br>Carlos D Aizenman |
| American Physiological Society | | Oscar A Carrillo<br>Carolina M Ramirez-Vizcarrondo |
| Brown University | | Oscar A Carrillo<br>Julia Bleier<br>Carolina M Ramirez-Vizcarrondo |
| NIH Office of the Director | NEI 5T32-EY018080 | Daniel L Felch |
| Bard Summer Research Institute, Bard College | | Molly McQuillan<br>Arseny S Khakhalin |

The funders had no role in study design, data collection and interpretation, or the decision to submit the work for publication.

### Author contributions

TLST, Conceptualization, Software, Formal analysis, Investigation, Visualization, Writing—original draft, Project administration, Writing—review and editing; OAC, Conceptualization, Formal analysis, Investigation, Visualization, Writing—original draft, Writing—review and editing; JB, Software, Formal analysis, Investigation, Visualization; CMR-V, Software, Investigation, Visualization, Methodology; DLF, MM, Formal analysis, Investigation, Methodology; CPT, Resources, Software, Formal analysis; ASK, Resources, Formal analysis, Investigation; CDA, Conceptualization, Formal analysis, Supervision, Funding acquisition, Writing—original draft, Project administration, Writing—review and editing

### Author ORCIDs

Torrey LS Truszkowski, http://orcid.org/0000-0001-9572-8851
Arseny S Khakhalin, http://orcid.org/0000-0002-0429-1728
Carlos D Aizenman, http://orcid.org/0000-0002-7378-7217

### Ethics

Animal experimentation: This study was performed in strict accordance with the recommendations in the Guide for the Care and Use of Laboratory Animals of the National Institutes of Health. All of the animals were handled according to approved institutional animal care and use committee (IACUC) protocols (1607000219) of Brown University.

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
