## [Decision Letter]

Thank you for submitting your article "A Cellular Mechanism for Inverse Effectiveness in Multisensory Integration" for consideration by *eLife*. Your article has been reviewed by two peer reviewers, and the evaluation has been overseen by Gary Westbrook as the Senior Editor and Reviewing Editor. The following individuals involved in review of your submission have agreed to reveal their identity: Barry Stein (Reviewer #1); Hollis T Cline (Reviewer #2).

Summary:

The reviewers have discussed the reviews with one another and the Senior Editor has drafted this decision to help you prepare a revised submission. As you will see from their comments below, both reviewers thought that this was an excellent manuscript and an appropriate Research Advance to complement your recent *eLife* paper. Please include a response to each comment in your revised cover letter and adjust the text of the manuscript to address each comment.

*Reviewer #1:*

This is a straightforward demonstration that a well-known property of multisensory integration ("Inverse Effectiveness") is present in the nonmammalian homologue of the Superior Colliculus – the Optic Tectum. The authors have done a nice job of using multiple techniques to demonstrate its presence, its operational constraints, its likely molecular basis, as well as its physiological and behavioral manifestations in *Xenopus* tadpoles. There are a few suggested changes for clarity.

1) It is interesting that the authors use the term "conserved" for the presence of Inverse Effectiveness. Presumably, they are assuming that the progenitor of amphibians and mammals had this property – although this is not stated. It is, of course, possible that it was invented and re-invented in different species. Also, the description in the text gives the impression that the authors think it has been conserved from mammalian forms. Clearly, this is not what they mean. One might say that it is actually conserved in more recent beasts (i.e., mammalians).

2) Including the behavior of the animal is really a good idea, because it shows that this physiological property is reflected in overt responses. It should be noted that a similar strategy had previously been used for similar reasons (see Stein et al. 1988, 1989 – cat behavior).

3) Is there confidence that the "acoustic" stimulus was activating the auditory system and not the vibratory (i.e., tactile) system (since the presence of the stimulus was judged by the vibration of the liquid in which the animals were swimming)?

4) Mechanosensory is the assumption the authors have for this stimulus – but, it is an electrical stimulus, not a mechanical one. As such, it could be activating any component of the somatosensory system – cutaneous, deep, nociceptive, etc. "Electrical stimulation of the skin" might a better descriptor.

5) Two different combinations of stimuli were used for testing different hypotheses (visual and "auditory", and visual (optic chiasm stimulation) and "mechanosensory". This was done for convenience and presumably there is no difference here in how the different modality combinations work, but a statement should be made somewhere as to why this was done (for convenience is fine).

*Reviewer #2:*

This paper examined the cellular basis for inverse effectiveness, a feature of multisensory integration, using *Xenopus* tapdoles. The authors use behavior, electrophysiological recordings and calcium imaging to examine the mechanisms underlying inverse effectiveness. They demonstrate that pairing auditory startle stimulus with weak visual input produces a stronger behavioral output (change in swim speed) that pairing the auditory input with weak stimulus, indicative of multisensory integration. They use calcium imaging to characterize neurons that are multisensory and show inverse effectiveness. They show that inverse effectiveness is not blocked by picrotoxin but is blocked by NMDA R. This study is beautifully conducted, and provides important cellular level resolution of the mechanisms underlying an important aspect of multisensory integration. I find no flaws with the manuscript, and only a single comment as below:

1) There is no need to abbreviate inverse effectiveness.

---

## [Author Response]

*[…] Reviewer #1:*

*This is a straightforward demonstration that a well-known property of multisensory integration ("Inverse Effectiveness") is present in the nonmammalian homologue of the Superior Colliculus – the Optic Tectum. The authors have done a nice job of using multiple techniques to demonstrate its presence, its operational constraints, its likely molecular basis, as well as its physiological and behavioral manifestations in Xenopus tadpoles. There are a few suggested changes for clarity.*

*1) It is interesting that the authors use the term "conserved" for the presence of Inverse Effectiveness. Presumably, they are assuming that the progenitor of amphibians and mammals had this property – although this is not stated. It is, of course, possible that it was invented and re-invented in different species. Also, the description in the text gives the impression that the authors think it has been conserved from mammalian forms. Clearly, this is not what they mean. One might say that it is actually conserved in more recent beasts (i.e., mammalians).*

We have altered the text to clarify this issue. We mean that this property and others described in prior paper are conserved across these vertebrate species. In particular we point to the Results and Discussion:

“These results extend those in our previous study to show that another important property of MSI is also present in the tectum, indicating that it is an evolutionarily important process conserved across vertebrates.”

*2) Including the behavior of the animal is really a good idea, because it shows that this physiological property is reflected in overt responses. It should be noted that a similar strategy had previously been used for similar reasons (see Stein et al. 1988, 1989 – cat behavior).*

Thank you for these references, they have been included in our Introduction:

“The phenomenological aspects of inverse effectiveness and other properties of MSI are well described in the vertebrate optic tectum or superior colliculus (Stein and Stanford 2008; Stein, Stanford, and Rowland 2014; Meredith and Stein 1983; Meredith and Stein 1986), but less is known about the cellular mechanisms underlying these processes or how these responses give rise to behavior (Stein, Huneycutt, and Meredith 1988; Stein et al. 1989). “

*3) Is there confidence that the "acoustic" stimulus was activating the auditory system and not the vibratory (i.e., tactile) system (since the presence of the stimulus was judged by the vibration of the liquid in which the animals were swimming)?*

This is a valid point and why we use the term mechanosensory to describe overall responses to acoustic stimuli. It is very difficult to prove whether the underwater stimulus is activating auditory or vibratory systems or both. We have clarified this in our text:

“The acoustic stimulus likely activates both the ear, skin and lateral line sensory systems.”

*4) Mechanosensory is the assumption the authors have for this stimulus – but, it is an electrical stimulus, not a mechanical one. As such, it could be activating any component of the somatosensory system – cutaneous, deep, nociceptive, etc. "Electrical stimulation of the skin" might a better descriptor.*

We have altered the text to reflect this:

“Trials were presented in blocks of 3 presentations of 4 stimuli conditions (multisensory, visual, electrical stimulation of the skin (mechanosensory) and no stimulus) […] The skin stimulation protocol is treated as functionally equivalent to presentation of acoustic stimuli (i.e. non-visual) and so was used for convenience in order to minimize mechanical movement of the preparation induced by the stimulus during Ca++ imaging.”

*5) Two different combinations of stimuli were used for testing different hypotheses (visual and "auditory", and visual (optic chiasm stimulation) and "mechanosensory". This was done for convenience and presumably there is no difference here in how the different modality combinations work, but a statement should be made somewhere as to why this was done (for convenience is fine).*

See reply to previous comment for changes made to the text. The main reason for using different stimuli is that during in vivo imaging, any mechanical stimuli would also move the very small preparation, making it hard to either record or image.

*Reviewer #2:*

*This paper examined the cellular basis for inverse effectiveness, a feature of multisensory integration, using Xenopus tapdoles. The authors use behavior, electrophysiological recordings and calcium imaging to examine the mechanisms underlying inverse effectiveness. They demonstrate that pairing auditory startle stimulus with weak visual input produces a stronger behavioral output (change in swim speed) that pairing the auditory input with weak stimulus, indicative of multisensory integration. They use calcium imaging to characterize neurons that are multisensory and show inverse effectiveness. They show that inverse effectiveness is not blocked by picrotoxin but is blocked by NMDA R. This study is beautifully conducted, and provides important cellular level resolution of the mechanisms underlying an important aspect of multisensory integration. I find no flaws with the manuscript, and only a single comment as below:*

*1) There is no need to abbreviate inverse effectiveness.*

Done.